# Clinical Characteristics, Differential Diagnosis and Genetic Analysis of Concentric Retinitis Pigmentosa

**DOI:** 10.3390/life11030260

**Published:** 2021-03-22

**Authors:** Mei Nakahara, Akio Oishi, Manabu Miyata, Hanako Ohashi Ikeda, Tomoko Hasegawa, Shogo Numa, Yuki Otsuka, Maho Oishi, Fumihiko Matsuda, Akitaka Tsujikawa

**Affiliations:** 1Department of Ophthalmology and Visual Sciences, Kyoto University Graduate School of Medicine, Kyoto 606-8507, Japan; mei_1012@kuhp.kyoto-u.ac.jp (M.N.); miyatam@kuhp.kyoto-u.ac.jp (M.M.); hanakoi@kuhp.kyoto-u.ac.jp (H.O.I.); tomoko11@kuhp.kyoto-u.ac.jp (T.H.); numa1988@kuhp.kyoto-u.ac.jp (S.N.); yotsuka@kuhp.kyoto-u.ac.jp (Y.O.); mah0ham@kuhp.kyoto-u.ac.jp (M.O.); tujikawa@kuhp.kyoto-u.ac.jp (A.T.); 2Department of Ophthalmology and Visual Sciences, Graduate School of Biomedical Sciences, Nagasaki University, Nagasaki 852-8504, Japan; 3Center for Genomic Medicine, Kyoto University Graduate School of Medicine, Kyoto 606-8507, Japan; fumi@genome.med.kyoto-u.ac.jp

**Keywords:** retinal dystrophy, retinitis pigmentosa, concentric retinitis pigmentosa, myotonic dystrophy, fundus autofluorescence

## Abstract

Concentric retinitis pigmentosa (RP), in which retinal degeneration is limited in the periphery, is rare and little information exists to date on the subject. Herein, we describe the clinical and genetic characteristics of this atypical form of RP. We retrospectively reviewed our database and identified 14 patients with concentric RP. Additionally, 14 patients with age-matched typical RP were also included. Patients with concentric RP had better visual acuity (logarithm of minimum angle of resolution −0.04 vs. 0.32, *p* = 0.047) and preserved ellipsoid zones (7630 µm vs. 2646 µm, *p* < 0.001) compared to typical RP. The electroretinogram showed subnormal but recordable responses in patients with concentric RP. Genetic testing was done in nine patients with concentric RP and revealed causative mutations in the *EYS* gene in one patient and the *RP9* gene in one patient. Two patients had myotonic dystrophy and the diagnosis was revised as myotonic dystrophy-associated retinopathy. Concentric RP is a rare, atypical form of RP with better visual function. There is some overlap in the causative genes in concentric and typical RP. Myotonic dystrophy-associated retinopathy is an important differential diagnosis.

## 1. Introduction

Inherited retinal dystrophy (IRD) includes a heterogenous group of diseases characterized by retinal cell death/impairment due to genetic causes. The most common form is retinitis pigmentosa (RP), which is a major cause of visual impairment in developed countries [1]. The classic triad of RP is bony spicule pigmentation, vessel attenuation and waxy pallor optic disc. The retinal degeneration generally starts with the mid-peripheral retina, inducing ring-shaped or concentric visual field loss, and progresses toward the macular and the periphery.

Ultra-wide-field scanning laser ophthalmoscopy is one of recent developments of imaging techniques in ophthalmology. It enables the observation of the peripheral retina with a single image [2]. Fundus autofluorescence (FAF) imaging is another imaging technique which visualizes retinal intrinsic fluorescent material. The degenerated retina and pigment epithelium generally show decreased FAF, and these are clearly depicted with the imaging [3]. A combination of these techniques, ultra-wide-field FAF imaging, is useful for monitoring patients with RP and provides otherwise inaccessible information [4].

Since we started routine examinations with peripheral FAF imaging, we noticed some cases with retinal degeneration limited to the peripheral retina. The clinical appearance reminded us of concentric RP, which was first described in a paper investigating visual field progression [5], and named later by Jacobson’s group [6]. They reported clinicopathologic pathologic finding of the disease; however, little was added thereafter. Autosomal dominant vitreoretinochoroidopathy (ADVIRC) is also characterized by a peripheral and circumferential retinal band of pigmentary alterations, and often accompanies microcornea [7]. Mutations of *BEST1* were identified as a cause of the disease [8]. In the present study, we describe multimodal imaging phenotypes and the genetic analysis of concentric RP in comparison with typical RP. In addition, another differential diagnosis, myotonic dystrophy-associated retinopathy, is discussed.

## 2. Materials and Methods

This retrospective study was approved by the ethics committee of the Kyoto University Graduate School of Medicine (Kyoto, Japan, study ID: R2198). The research followed the tenets of the Declaration of Helsinki. Some of the patients also participated in genetic testing study, which was approved by the same ethics committee (G0746-8), and the participants provided written informed consent.

We reviewed the in-house database of patients who visited the retinal dystrophy clinic of Kyoto University Hospital between January 2012 and December 2019. To extract the specific case series of concentric RP, we defined following criteria: (1) bilaterally symmetrical peripheral retinal degeneration with or without bony spicule pigmentation; (2) the four-disc-diameter area centering the fovea is spared on FAF image (Figure 1); (3) absence of other acquired retinal diseases, particularly age-related macular degeneration; (4) no history of intraocular surgery other than cataracts. The age of onset and initial symptoms were recorded from clinical interviews.

To highlight the distinct clinical characteristics of the patients, we recruited age-matched patients with typical RP. A patient with the closest birthday was selected for each patient with concentric RP. We investigated visual acuity, measured with the Landolt chart and converted to the logarithm of minimum angle of resolution (LogMAR), a mean deviation and a central threshold of the Humphrey visual field analyzer 10-2 program. Horizontal and vertical scans through the fovea were obtained with spectral-domain optical coherence tomography (Spectralis, Heidelberg Engineering, Heidelberg, Germany). The central retinal thickness was measured from inner surface of the internal limiting membrane to the inner surface of the retinal pigment epithelium, and the width of the ellipsoid zone was measured on the horizontal and vertical images and averaged for each patient. The measurements for the right eye were compared between concentric RP and typical RP using an unpaired t-test. Data are presented as mean ± standard deviation. The Snellen equivalent is presented for visual acuity for easier understanding. Goldmann kinetic visual field tests were reviewed when available.

Genetic testing was done as previously reported [9,10,11].

## 3. Results

There were 1776 patients with IRD including 673 with RP during the study period. We identified 15 patients from 14 families with concentric RP who met all the inclusion criteria, but 1 patient was excluded because the patient was accompanied with central serous chorioretinopathy at the initial presentation. Thus, we analyzed 14 cases from 13 families with concentric RP. As the prevalence indicates, the criteria successfully differentiated typical RP. Additionally, 14 age-matched patients from 14 families with typical RP were also included. Details of all the patients are shown in Table 1 and a representative case is shown in Figure 2.

The mean age of the concentric RP patients was 61.3 ± 17.6 and five patient (26.3%) were men. Patients #1 and #2, who are siblings, reported that their father was diagnosed with some retinal dystrophy, but the details were unavailable. The other cases were sporadic. Two unrelated cases (Patients #4 and #5) had myotonic dystrophy as a comorbidity and were considered as myotonic dystrophy-associated retinopathy (Figure 3). None reported the use of chlorpromazine, thioridazine or hydroxychloroquine. None reported an episode of acute or subacute vision loss, indicating the absence of past uveitis or acute zonal occult outer retinopathy. The corneal diameter was measured in seven patients and showed normal values (>11 mm). None of the patients had a cystoid macular edema. Patient #13 developed a macular hole during the follow-up visit. Goldmann kinetic perimetry was done in seven patients and showed concentric visual field constriction corresponding to the peripheral degeneration.

Table 2 shows comparison of clinical characteristics of concentric RP and typical RP. As expected, patients with concentric RP had better visual acuity, central visual field, central threshold, photopic and scotopic electroretinogram (ERG) amplitude and a wider ellipsoid zone. The representative ERG is shown in Figure 4. The central threshold also looked larger, but there was no statistical significance. The central retinal thickness was not different between the two groups, partly due to the cases with cystoid macular edema and epiretinal membrane-induced retinal thickening being included in the typical RP group.

Eight patients with concentric RP underwent direct sequencing of *BEST1* and eight underwent exome sequencing (DNA sample was unavailable in five patients). The screening revealed a homozygous *EYS* mutation in one patient and a heterozygous *RP9* mutation in another patient. The identified variants are shown in Table 1. Exome sequencing was done for 13 patients in the typical RP group, and revealed *EYS* (2 patients, c.4957dupA homozygous and c.4957dupA/c.8805C>A), *RHO* (1 patient, c.44A>G), *CEP164* (1 patient, c.452G>A homozygous) and *CNGA1* (1 patient, c.398delG/c.1196A>G) as causative genes.

## 4. Discussion

In the present paper, we describe the characteristics of patients with concentric RP who exhibited retinal degeneration only in the periphery. Two of the included patients were later diagnosed with myotonic dystrophy-associated retinopathy. Two patients showed causative mutations in RP-associated genes. The report provides representative figures for the long-unattended phenotype.

Patients with concentric RP showed some distinct features compared to typical RP. Firstly, as defined by the inclusion criteria, the four-disc-diameter retina was spared. This is very rare in typical RP, even in the early stages. The area is generally affected, even in another mild phenotype sectoral RP [12]. Although the age of onset was not specified in all cases, more than half of the cases were patients older than 60 years, indicating that the disease progresses slowly, or that the onset is late compared to typical RP. Corresponding to the spared retina, the ERG was recordable in all the examined patients, even those in their 60s. This is also uncommon in typical RP. Meanwhile, genetic testing revealed mutations of RP-associated genes in some patients. The result indicates that these patients, at least those with identified causative mutations in the RP gene, have an atypical mild phenotype of RP compared to sectoral RP [12,13]. The original report also discussed the presumed X factor that induced retinal degeneration from the periphery, but it has yet to be elucidated [6]. Investigating what makes peripheral degeneration occur and why these patients retain the mild phenotype might lead to identifying modifiable factors for the disease.

The involvement of the peripheral retina with macular sparing suggests ADVIRC as a differential diagnosis. The disease is characterized by a peripheral circumferential retinal band of pigmentary alterations [7]. Mutations of *BEST1* were identified as a cause of the disease [8]. Indeed, the present cases showed similar pigmentary degeneration in the peripheral retina. However, none of the investigated patients had mutations in *BEST1*. Microcornea, which is common in ADVIRC, was not observed either. Vitreoretinal dystrophy, such as Stickler syndrome and Wagner syndrome, also affects the peripheral retina [14,15], but it is less likely, considering the lack of systemic symptoms, no causative mutation in the corresponding genes and the normal appearance of the vitreous.

Peripheral abnormality has also been seen in age-related macular degeneration (AMD) or in elderly patients [16,17,18]. We could not totally differentiate the senile reticular pigmentary changes, especially in the elderly cases, despite the fact that we excluded cases with AMD features. However, it is unlikely that such age-related changes would already occur in a patient’s 20s. The reduced amplitude of ERG indicates pan retinal dysfunction, which may not be present in senile peripheral changes. In addition, the present cases tended to show a clear border and more prominent pigmentary changes than the reported age-related changes.

Two cases had myotonic dystrophy in the cohort. It is known that myotonic dystrophy may accompany peripheral atrophic and/or pigmentary changes [19,20,21]. However, the peripheral change is less focused compared to the butterfly-shaped or reticular macular changes seen in the disease [22], The present cases can be representative examples on how the peripheral lesion appears in wide-field FAF images. Care should be taken for such unusual features, because ophthalmic findings may be the earliest symptom of inherited neurodegenerative diseases [23].

In conclusion, we reported characteristics of concentric RP in which retinal degeneration was limited to the periphery. Myotonic dystrophy-associated retinopathy as well as ADVIRC were the important differential diagnoses. We assume that many patients are undetected or left undiagnosed due to the mild symptom. Further studies are warranted for determining the prevalence, clinical course and difference of concentric RP from typical RP.

## Figures and Tables

**Figure 1 life-11-00260-f001:**
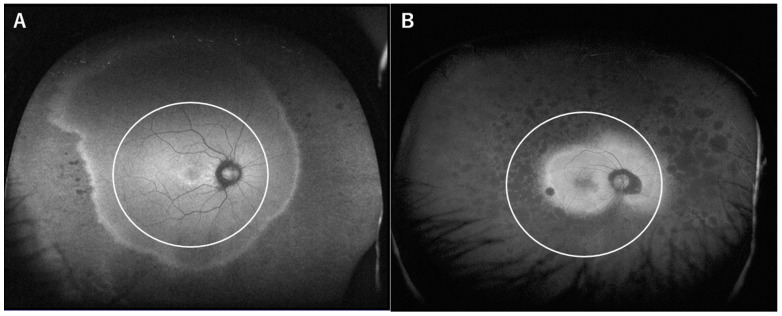
Wide-field fundus autofluorescent images of concentric retinitis pigmentosa (RP) (**A**) and typical RP (**B**). In the present study, concentric RP was defined as the preservation of at least a four-disc-diameter area from the fovea. As shown in panel B, this criterion excluded virtually all the typical RP cases.

**Figure 2 life-11-00260-f002:**
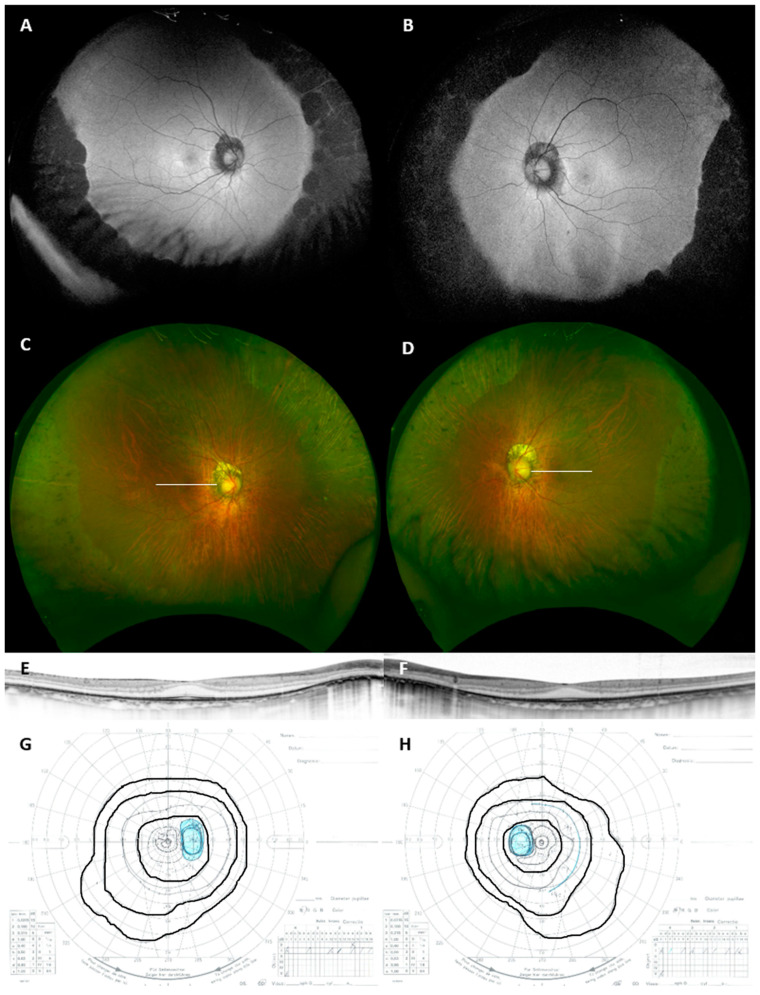
Wide-field fundus autofluorescent (**A**,**B**), pseudo-color (**C**,**D**), optical coherence tomography (OCT; (**E**,**F**)) and Goldmann kinetic perimetry (**G**,**H**) images of a representative case of concentric RP (Patient #3). OCT scan lines are shown in panels C and D. V/4e, I/4e and I/2e isopters are traced in panels G and H. Note the symmetric and well-demarcated peripheral retinal degeneration. The outer nuclear layer and ellipsoid zone were relatively well preserved despite focal thinning and impairment. Mild concentric constriction of the visual field was noted.

**Figure 3 life-11-00260-f003:**
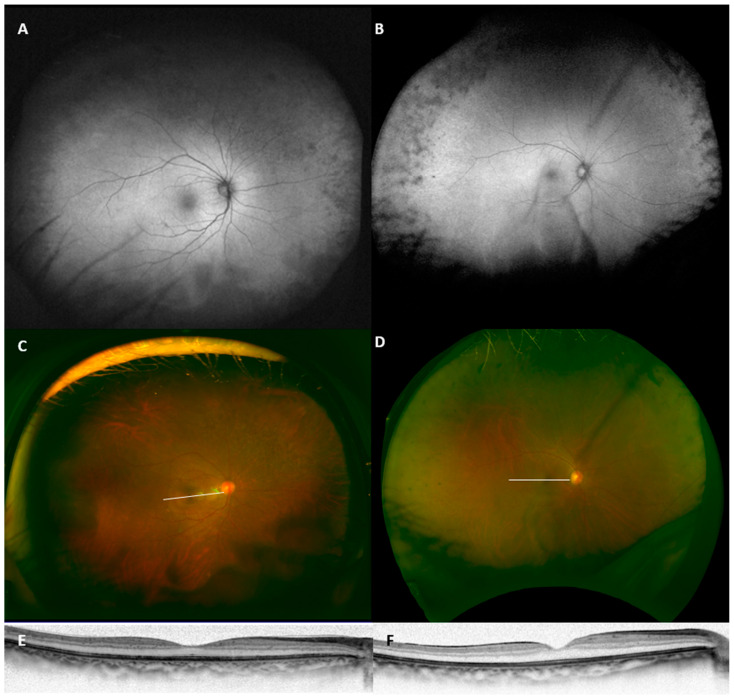
Wide-field fundus autofluorescent (**A**,**B**), pseudo-color (**C**,**D**) and optical coherence tomography (**E**,**F**) images of two cases of myotonic dystrophy-associated retinopathy (Patients #4 and #5).

**Figure 4 life-11-00260-f004:**
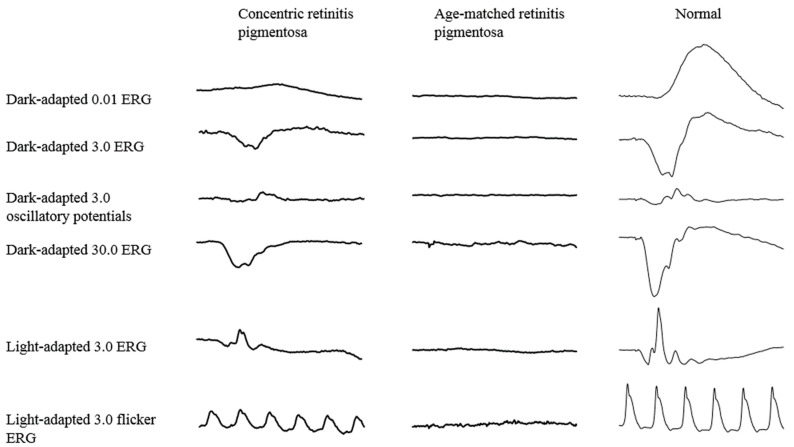
Representative electroretinogram (ERG) waveforms of patients with concentric retinitis pigmentosa (Patient #9), age-matched retinitis pimentosa and a normal subject. The concentric retinitis pigmentosa patient still showed reduced but recordable ERG in his 60s.

**Table 1 life-11-00260-t001:** Clinical characteristics of patients with concentric RP included in the study.

	Sex	Age	Onset Age	Symptom	Visual Acuity	Scotopic ERG	Photopic ERG	Detected Mutation	Comorbidities
#1 *	Woman	29	22	photopsia	1.2	normal	normal	none	Renal dysfunction
#2 *	Woman	36	28	blurred vision at evening	1.2	non recordable	normal	none	
#3	Woman	41	diagnosed at 38	none	1.2	unavailable	unavailable	unavailable	Interstitial pneumonia
#4	Woman	47	unclear	unclear	0.8	reduced amplitude	reduced amplitude	none	Myotonic dystrophy, Diabetes
#5	Woman	50	20s	blurred vision and floaters	1.2	unavailable	unavailable	unavailable	Myotonic dystrophy
#6	Man	64	58	blurred vision at evening	1.0	reduced amplitude	reduced amplitude	none	Intervertebral disc hernia
#7	Man	66	unclear	unclear	1.2	reduced amplitude	reduced amplitude	unavailable	Angina pectoris
#8	Woman	69	diagnosed at 62	none	1.2	unavailable	unavailable	*EYS* (c.2528G>A: p.G843E, homozygous)	
#9	Man	70	40s	nyctalopia	1.5	reduced amplitude	normal	none	
#10	Woman	71	diagnosed at 60	none	1.0	reduced amplitude	reduced amplitude	unavailable	Thyroid gland cancer
#11	Woman	74	50s	nyctalopia	1.2	reduced amplitude	reduced amplitude	none	Diabetes
#12	Woman	75	58	stumbling (visual field restriction)	0.9	reduced amplitude	reduced amplitude	none	
#13	Man	79	40s	nyctalopia	0.9	reduced amplitude	reduced amplitude	none	
#14	Man	87	diagnosed at 66	none	1.0	reduced amplitude	reduced amplitude	*RP9* (c.509A>G: p.D170G)	Diabetes, Coronary artery occlusion

* #1 and #2 are siblings; visual acuity and decimal visual acuity measured with Landolt C charts; ERG, electroretinogram.

**Table 2 life-11-00260-t002:** Comparison between concentric retinitis pigmentosa and typical retinitis pigmentosa.

	Concentric RP (*n* = 14)	Typical RP (*n* = 14)	*p*-Value	Reference In-House Normal Range
Age (years)	61.3 ± 17.6	61.4 ± 17.1	0.99	
Sex (Man/Woman)	5/9	5/9	1.00	
Visual acuity (logMAR)	−0.04 ± 0.07	0.32 ± 0.60	0.047	
HFA 10-2 MD (dB)	−3.1 ± 3.0	−18.1 ± 11.8	<0.001	
Central threshold (dB)	33.7 ± 2.1	27.8 ± 9.6	0.04	
Dark adapted 0.01 ERG b-wave (µV)	28.9 ± 20.5	6.3 ± 11.9	0.002	81–128
Dark adapted 3.0 ERG a-wave (µV)	55.9 ± 34.6	15.0 ± 17.4	0.003	67–281
Dark adapted 3.0 ERG b-wave (µV)	102.1 ± 39.2	23.5 ± 23.3	<0.001	139–361
Light adapted 3.0 ERG b-wave (µV)	59.0 ± 23.8	12.1 ± 9.9	<0.001	83–259
Central retinal thickness (µm)	215.6 ± 42.0	218.8 ± 116.0	0.92	
Ellipsoid zone width (µm)	7630 ± 1284	2646 ± 2488	<0.001	

RP, retinitis pigmentosa; logMAR, logarithm of minimum angle of resolution; HFA, Humphrey visual field analyzer; MD, mean deviation; ERG, electroretinogram.

## Data Availability

All the relevant data are presented. Other data are available from the corresponding author on reasonable request.

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
