# Peer review of "Clinical Characteristics, Differential Diagnosis and Genetic Analysis of Concentric Retinitis Pigmentosa"

_life, 2021, doi:10.3390/life11030260_

Round 1
Reviewer 1 Report
A very interesting paper about a rare clinical entity. Congratulations.
Just one comment. In the text you mention that 5 patients were male. However, in the table one can count 6 male patients. Please clarify.
Author Response
We appreciate your positive feedback and careful reading. We found that there was a mistake in the table; Patient #10 was female. The table was corrected.
Reviewer 2 Report
In this paper the authors describe clinical and genetic characteristics of concentric Retinitis Pigmentosa(RP). In the present study, they describe multimodal imaging phenotypes and genetic analysis of concentric RP in comparison with typical RP. They use a data base of 14 concentric RP patients and an age matched typical RP group. The investigation of these groups revealed an overlap in causative genes in concentric and typical RP (despite the lack of phenotypes in the concentric RP).
This research can be of value for better diagnosing the currently undertreated and underdiagnosed cases of concentric RP. Notwithstanding the importance of this work and the contributions it may have clinically, the cohort and the population size utilized to draw these conclusions is not sufficiently large. Moreover, as the authors note the study lacks the understanding of the difference between concentric and typical RP and further study is needed. Specifically, the authors note that “Searching for novel causative genes that cause only mild phenotype would be another interesting theme to pursue.” Can the authors on how they intend to pursue this theme? And what other genes they aim at investigating?
I recommend some revision before this paper can be published
- A larger population
- Represent genetic data(missing from this report) -although they refer to a different publication which describes the sequencing , this paper would greatly benefit if this information is added
Minor: This paper has some minor grammatical mistakes.
Author Response
Thank you for your educational comment. As the reviewer pointed out, the result imply that causative genes overlap between concentric and typical RP. We do not have specific gene in mind and do not have any plan to search novel genes. We deleted the sentence.
We agree that sample size is relatively small, but still consider that even the report of 14 cases may of value in attracting attention to this less recognized atypical form of RP. Considering that the phenotype is very rare (as stated in the first sentence of the result, only 14 families were identified among 1776 IRD cases), it is unrealistic to expand the cohort.
The information of the identified variants for concentric RP was shown in Table 1. We added the information for typical RP.
We are sorry for our English despite that the paper underwent professional language proof. We carefully checked the paper again.